# Expression of Immune-Related and Inflammatory Markers and Their Prognostic Impact in Colorectal Cancer Patients

**DOI:** 10.3390/ijms241411579

**Published:** 2023-07-18

**Authors:** Sanghyun An, Soo-Ki Kim, Hye Youn Kwon, Cheol Su Kim, Hui-Jae Bang, Hyejin Do, BoRa Kim, Kwangmin Kim, Youngwan Kim

**Affiliations:** 1Department of Colorectal Surgery, Yonsei University Wonju College of Medicine, Wonju 26426, Republic of Korea; uldura@yonsei.ac.kr (S.A.); kwonhy@yonsei.ac.kr (H.Y.K.); 2Wonju Surgical Research Collaboration, Wonju 26465, Republic of Korea; banghuijae@yonsei.ac.kr; 3Department of Microbiology, Yonsei University Wonju College of Medicine, Wonju 26426, Republic of Korea; kim6@yonsei.ac.kr (S.-K.K.); cs-kim@yonsei.ac.kr (C.S.K.); 4Department of Surgery, Yongin Severance Hospital, Yonsei University College of Medicine, Yongin 16995, Republic of Korea; 5Department of Anesthesiology, Yonsei University Wonju College of Medicine, Wonju 26426, Republic of Korea; dohyejin@yonsei.ac.kr; 6Department of Internal Medicine, Division of Gastroenterology, Yonsei University Wonju College of Medicine, Wonju 26426, Republic of Korea; md05bkim@yonsei.ac.kr

**Keywords:** colorectal neoplasm, biomarkers, bioplex, immune checkpoint proteins

## Abstract

The tumor microenvironment of colorectal cancer (CRC) is heterogenous; thus, it is likely that multiple immune-related and inflammatory markers are simultaneously expressed in the tumor. The aim of this study was to identify immune-related and inflammatory markers expressed in freshly frozen CRC tissues and to investigate whether they are related to the clinicopathological features and prognosis of CRC. Seventy patients with CRC who underwent curative surgical resection between December 2014 and January 2017 were included in this study. Tissue samples were obtained from tumor and non-tumor areas in the patients’ colons. The concentrations of immune-related markers (APRIL/TNFSF13, BAFF, LAG-3, PD-1, PD-L1, and CTLA-4) and inflammatory markers (CHIT, MMP-3, osteocalcin, pentraxin-3, sTNF-R1, and sTNF-R2) in the samples were measured using the Bio-plex Multiplex Immunoassay system. The concentrations of APRIL/TNFSF13, BAFF, and MMP-3 in the samples were significantly high; thus, we conducted analyses based on the cut-off values for these three markers. The high-APRIL/TNFSH13-expression group showed a significantly higher rate of metastatic lesions than the low-expression group, whereas the high-MMP-3-expression group had higher CEA levels, more lymph node metastases, and more advanced disease stages than the low-expression group. The five-year disease-free survival of the high-MMP-3-expression group was significantly shorter than that of the low-expression group (65.1% vs. 90.2%, *p* = 0.033). This study provides evidence that the APRIL/TNFSF13, BAFF, and MMP-3 pathway is overexpressed in CRC tissues and is associated with unfavorable clinicopathological features and poor prognosis in CRC patients. These markers could serve as diagnostic or prognostic biomarkers for CRC.

## 1. Introduction

In recent years, immunotherapy using immune checkpoint inhibitors (ICIs) has been receiving increasing attention as an effective therapeutic option for the treatment of advanced, metastatic, and recurrent colorectal cancer (CRC) [1,2]. Immunotherapy using ICIs can reverse tumor immune escape by suppressing immune checkpoint pathways. Representative immune checkpoint target molecules include programmed cell death protein 1 (PD-1), programmed death ligand 1 (PD-L1), and cytotoxic T-lymphocyte-associated protein 4 (CTLA-4). The efficacy of ICI therapy for the treatment of various malignant tumors, such as melanoma, lung cancer, and prostate cancer, has been demonstrated [3,4,5,6]. For the treatment of CRC, the ICI therapy has been partially proven to be effective and is used in clinical settings. Currently, ICIs are used on a limited basis for the treatment of patients with stage IV metastatic CRC and have been reported to have some effect in patients with the DNA mismatch repair-deficient (dMMR)/microsatellite instability-high (MSI-H) genetic phenotype. However, some patients do not respond to immunotherapy; therefore, its effect has not been clearly identified [7,8]. Additionally, ICIs are ineffective for the treatment of tumors with specific genetic phenotypes, such as the mismatch repair-proficient (pMMR) and microsatellite-stable (MSS) phenotypes, or with low levels of microsatellite instability (MSI-L) [9]. As tumors with the MSI-H phenotype account for only 5–15% of all CRC cases, the development of immunotherapy for pMMR/MSS tumors with few genetic variants is urgently needed [10]. In recent years, there has been research focused on identifying and targeting additional immune checkpoints in the tumor microenvironment to enhance the effectiveness of immunotherapy. Immune checkpoints, including lymphocyte activation gene-3 (LAG-3), T cell immunoglobulin and mucin domain 3 (TIM-3), B7-homolog 3 (B7-H3), V-domain immunoglobulin-containing suppressor of T cell activation (VISTA), diacylglycerol kinase-α (DGK-α), T cell immunoglobulin and ITIM domain (TIGIT), and B and T lymphocyte attenuator (BTLA), have gained attention as potential targets for immunotherapy [11]. Cytotoxic lymphocytes, particularly, cytotoxic T lymphocytes (CTLs), are considered crucial components of the immune system’s anti-tumor response. CTLs play a vital role in recognizing and eliminating cancer cells [12]. To improve CRC immunotherapy, it is essential to discover not only the MSI status but also novel immune-related markers that can predict the efficacy of immunotherapy for cancer treatment. The overexpression of various immune-related markers in tumors has been reported in several studies. However, to the best of our knowledge, there has been no clear conclusion that these markers are related to clinicopathological features and long-term outcomes [13,14,15].

Given the heterogeneous nature of the tumor microenvironment (TME), it is likely that multiple immune-related and inflammatory markers are simultaneously expressed in tumors, contributing to the complexity of tumor–immune interactions. Understanding the expression patterns and effects of these markers in the TME is critical for developing effective cancer therapies that harness the power of the immune system to fight cancer [16].

Therefore, this study aimed to identify immune-related and inflammatory markers expressed in CRC tissue samples that had been stored as freshly frozen samples and to evaluate whether the identified markers are related to clinicopathological features and prognosis. By examining the relationship between the immune-related and inflammatory markers and patient prognosis, we sought to determine whether these markers could serve as prognostic indicators for CRC patients. In particular, rather than use a traditional enzyme-linked immunosorbent assay (ELISA), we attempted to efficiently identify these markers using the Bio-plex Multiplex immunoassay, which has not been used in previous studies.

## 2. Results

### 2.1. Patient Characteristics

Seventy patients were included in this study. The mean age of the patients was 69.6 years, and 38 (54.3%) of them were male. Nineteen (27.1%) and twenty-seven (25.7%) patients had right- and left-sided colon cancer, respectively, whereas twenty-four (34.3%) had rectal cancer. Fifty-five (78.6%) patients underwent minimally invasive surgery, including laparoscopic or robotic surgery. Thirty-one (44.3%) patients had stage III disease, which was the most common disease stage, and the average number of metastatic lymph nodes recorded was 2.2. Sixty-three (91%) patients had MSS (91%), four had MSI-H (5%), and three (4%) had unverifiable MSI information. Forty-six (65.7%) patients received postoperative chemotherapy, whereas only one (1.4%) patient received radiation treatment. During the follow-up period, 17 (24.3%) patients showed cancer recurrence, and 5 (7.1%) patients died. The patient characteristics are summarized in Table 1.

### 2.2. Immune-Related and Inflammatory Markers in the Tumor Tissues

Table 2 shows the results of the multiplex immunoassay for immune-related and inflammatory markers in the tumor tissues. Among the various markers identified in the tumor tissues, a proliferation-inducing ligand (APRIL/TNFSF13), B cell activating factor (BAFF), and matrix metalloproteinase-3 (MMP-3) showed significantly high concentrations. Scatter plots that show the distribution of the immune marker levels are outlined in Figure 1.

### 2.3. Relationships between Immune-Related and Inflammatory Markers and Clinicopathologic Features

We set cut-off values for each of the three above-mentioned immune-related and inflammatory markers expressed at significantly high levels in the tumor tissues and analyzed the differences in clinicopathological features between the low- and high-expression groups. APRIL/TNFSH13 and BAFF were grouped based on the quartile 3 values of 806.4 and 664.0, respectively, whereas MMP-3 was grouped based on the quartile 1 value of 736.2. There were no differences in the basic patient characteristics, such as age, body mass index, American Society of Anesthesiologists score, and medical history, between the subgroups. The high-APRIL/TNFSH13-expression group showed a significantly higher rate of metastatic lesions (ML) than the low-APRIL/TNFSH13-expression group (11.8% vs. 36.8%, *p* = 0.03); therefore, the proportion of patients with stage IV disease was high. Additionally, the neutrophil/lymphocyte ratio, one of the serologic markers, was significantly higher in the high-APRIL/TNFSH13-expression group than in the low-APRIL/TNFSH13-expression group (4.1 vs. 2.7, respectively, *p* = 0.04). The high-BAFF-expression group had a significantly higher five-year recurrence rate than the low-BAFF-expression group (12 [23.1%] vs. 5 [27.8%], *p* = 0.03). Compared to the low-expression group, the high-MMP-3-expression group had more patients with a CEA level ≥ 5 (44.7% vs. 17.4%, *p* = 0.04). In addition, the high-MMP-3-expression group had a higher proportion of patients with T4 disease than the low-MMP-3-expression group (15 [31.9%] vs. 1 [4.3%], *p* = 0.01). Furthermore, the mean number of metastatic lymph nodes in the high-MMP-3-expression group was 2.7, which was significantly higher than that of the low-MMP-3-expression group, which was 1.1 (*p* = 0.02), and there were many patients with advanced-stage disease, such as stage 3 and 4 disease (*p* = 0.04) (Table 3).

### 2.4. Relationships between Immune-Related and Inflammatory Markers and Long-Term Oncologic Outcomes

Of the three above-mentioned markers, only MMP-3 was associated with the five-year disease-free survival (DFS). The five-year DFS of the high-MMP-3-expression group was significantly lower than that of the low-MMP-3-expression group (65.1% vs. 90.2%, *p* = 0.033). In addition, there was no difference in the five-year overall survival (OS) between the two groups (90.0% vs. 95.7%, *p* = 0.489). For APRIL/TNFSH13 and BAFF, there was no difference between the survival curves of the high- and low-expression groups (Figure 2).

## 3. Discussion

In this study, we investigated the expression of immune-related and inflammatory markers in CRC tissue samples. The results indicated that APRIL/TNFSF13, BAFF, and MMP-3 were highly expressed in CRCs, and a high expression of immune-related and inflammatory markers was associated with advanced clinicopathological features. In our analysis, a high expression of MMP-3 was associated with elevated CEA levels, more lymph node metastases, and more advanced disease stages, and the MMP-3 expression level was associated with long-term prognosis, such as five-year DFS. In addition, the high-APRIL/TNFSH13-expression group showed a higher rate of metastatic lesions than the low-APRIL/TNFSH13-expression group.

MMPs are a family of at least 28 zinc-dependent enzymes. Their main function is catalyzing proteolytic activities and aiding the breakdown of the extracellular matrix. They are upregulated in response to inflammation and have been shown to be involved in the development and progression of several inflammatory and autoimmune diseases, as well as in cancer. They induce tumor invasion, neoangiogenesis, and metastasis by degrading the connective tissues between cells and in the lining of blood vessels [17]. Several studies demonstrated that MMPs are more highly expressed in malignant tumors compared with normal tissue [14,18,19]. There are several subtypes of MMPs, and studies on the role of each subtype have been conducted [18,20]. Yu et al. [18] analyzed the mRNA expression levels of all 24 MMPs and their prognostic values in CRC. The authors suggested that the transcriptional levels of MMP1, MMP3, MMP7, MMP9–MMP12, and MMP14 are significantly upregulated in tumors. In addition, their analysis showed that the upregulation of MMP11, MMP14, MMP17, and MMP19 is significantly associated with a more advanced tumor stage and a worse long-term prognosis. Another study by Islekel et al. [13] demonstrated that the protein expression levels of MMP-3 in tumor tissues are higher than those in normal tissues, and that the expression level of MMP-3 is related to the lymph node status. These findings are consistent with the results of the present study.

APRIL/TNFSF13 is a tumor necrosis factor (TNF) protein that plays an important role in the development of B cells, which are involved in immune function [21]. This protein is expressed by immune cells in the bone marrow and peripheral tissues under normal physiological conditions. APRIL is produced by various types of tumor cells, including breast, gastric, bladder, and ovarian cancer cells [22,23,24,25]. Several studies have suggested that APRIL is overexpressed in CRC tissues and that increased APRIL expression is associated with poor prognosis in patients with CRC [26,27,28]. Similar to the results of the present study, a study by Lascano et al. [26] demonstrated that distant metastasis is more frequent in patients that show high APRIL expression than in those that show low APRIL expression; however, the APRIL expression level is not an independent factor for OS.

BAFF is a member of the TNF superfamily and is mainly produced by myeloid cells. BAFF plays a role in the immune function, regulating B cell survival, activation, and maturation. Previous studies demonstrated that BAFF plays a role in neoplasm progression and aggressiveness [29,30]. In addition, both BAFF and APRIL signaling may increase tumor cell proliferation and enhance tumor cell viability in response to chemotherapeutic drugs for hematopoietic malignancies. Interestingly, elevated blood levels of BAFF and APRIL are associated with an advanced disease stage and invasiveness of cancers such as breast cancer, chronic lymphocytic leukemia, and pancreatic cancer [30,31]. However, the relationship between BAFF expression and disease progression is not consistent in all cancers [32]. A study showed that the expression of BAFF in CRC tissues is higher than that in normal tissues; however, there has been no analysis of the association between BAFF expression and disease severity [15]. Another study demonstrated the markers APRIL and BAFF did not show any correlation with the tumor stage. However, they exhibited an inverse relationship with the immune infiltrate level and CD163 tissue expression [15]. In the present study, high BAFF expression was observed; however, its correlations with clinicopathological features and long-term outcomes were not confirmed.

Taken together, the APRIL/TNFSF13, BAFF/TNFSF13B, and MMP-3 pathway plays a critical role in the regulation of the immune system, and dysregulation of this pathway can lead to the development of cancer. Theoretically, these above-mentioned markers that were overexpressed in CRC tissues in the present study can be used as treatment targets. In fact, several studies have been conducted to evaluate the potential of these markers as therapeutic targets. MMP-3 has also been studied as a potential therapeutic target for CRC [33]. The results of an in vitro study by Wen et al. suggested that histone deacetylase 11 (HDAC11) inhibits the migration and invasion of CRC cells by downregulating MMP-3 expression. In the study, the authors found that HDAC11, a member of the histone deacetylase (HDAC) family, is downregulated in human colorectal cancer (CRC) tissues. They observed a correlation between decreased HDAC11 levels and advanced clinical stage as well as lymph node metastasis in CRC patients. Another in vivo and in vitro study suggested that APRIL knockdown is associated with the modulation of cell proliferation, as well as with the reduction of cell migration and invasion in vitro. Moreover, APRIL-knockdown cells displayed markedly inhibited tumor growth and decreased liver metastasis in the study. The study also revealed that APRIL could regulate the expression of MMPs, suggesting a link between APRIL and MMPs [34]. Overall, these studies suggest that APRIL/TNFSF13 and MMP-3 may be potential targets for the development of new therapies for CRC. However, further research is required to fully understand the roles of these markers in the development and progression of CRC.

To assess the expression of immune-related and inflammatory markers in the present study, we utilized the Bio-plex Multiplex immunoassay system. This is a highly sensitive assay that is capable of simultaneously measuring multiple analytes in a single sample, which makes it an ideal tool for studying complex biological systems such as the TME [35]. The multiplex immunoassay is faster than the traditional ELISA and planar microarray, but with comparable and reliable diagnostic accuracy [36,37,38]. The multiplex immunoassay has been used in a few studies to detect tumor-specific biomarkers in malignant tumors such as melanoma, ovarian cancer, and pancreatic cancer [39,40], and Calu et al. [15] used multiplex immunoassays to identify inflammatory molecules in CRC tissues.

In our research, well-known immune checkpoint molecules such as PD-1, PD-L1, CTLA-4, and LAG-3 were not highly expressed in the tumor tissues. Additionally, we did not observe significant expression of CHIT, pentraxin-3, sTNF-R1, sTNF-R2, and osteocalcin. However, some articles reported the expression of these immune and inflammatory-related markers in tumor cells and their implications in tumor biology, contrary to our findings [15,41,42,43,44,45,46]. Lee et al. [42] performed immunohistochemistry to measure the expression of immune markers in tumor cells in a cohort of 395 colorectal cancer patients. They reported that high expression of PD-L1 was observed in 5% of the tumor cells, whereas PD-1 exhibited high expression in 19% of tumor-infiltrating lymphocytes, and in their analysis, PD-L1 expression in tumor cells was confirmed as a poor prognostic factor for recurrence-free survival, while PD-1 expression in tumor-infiltrating lymphocytes was identified as a favorable prognostic factor. These discrepancies observed compared to other studies can be explained by differences in the research methodologies.

This study has several limitations. First, the number of patients included in this study was relatively small. Second, as this study was conducted using data from a single institution, the characteristics of various types of patients may not have been included. Third, since the absolute expression value or cut-off value of each marker has not been established so far, we set the cut-off values as representative values showing clinically meaningful results for each marker. Therefore, there may be limitations in generalizing and applying the cut-off values used in our research. Fourth, markers other than APRIL/TNFSF13, BAFF, and MMP-3 did not yield meaningful results because their expression was below our detection level; therefore, their effects could not be studied. Given that the TME is dynamic and subject to changes over time, we utilized freshly frozen tissue samples to capture the most current state of the tumors and their surrounding environment. It is possible that certain aspects of the microenvironment may not have been fully captured because the freeze–thaw process can alter the expression of certain biomolecules. However, the use of frozen tissue samples is a widely accepted method for studying gene expression and protein levels and has yielded reliable results in various research settings [47]. Notwithstanding these limitations, our study has a strong point in that it simultaneously examined the expression of immune checkpoint molecules and markers related to inflammation in tumor tissue. Based on these concepts, it is anticipated that future research, taking into account both immunity and inflammation, can explore therapeutic approaches targeting inflammatory markers in addition to immune checkpoint inhibitors.

## 4. Materials and Methods

### 4.1. Study Population

Patients with CRC who underwent curative surgical resection between December 2014 and January 2017 were included in this study. Patients who underwent bypass surgery without curative resection for palliative purposes, patients with cancers of other organs, and patients with a previous history of CRC were excluded from the study. Tumor tissues were prospectively obtained during the index surgical procedure. Clinical, hematological, and pathological information was extracted from the patients’ medical records. Informed consent to participate in this study was obtained from all patients. The study protocol was approved by the Institutional Review Board of Wonju Severance Medical Center (CR:318334), and the study was conducted in accordance with the tenets of the Declaration of Helsinki.

### 4.2. Tissue Sample Preparation

Tissue samples from both tumor and non-tumor areas of the resected specimen were collected and immediately placed in an ice-cold radioimmunoprecipitation assay buffer that contained protease inhibitor cocktails (Sigma Chemical Co., St. Louis, MO, USA). The tissue samples were homogenized at 14,000 rpm for 10 min and centrifuged for 5 min. The resultant supernatant was collected and stored at −80 °C until it was ready for further analysis.

### 4.3. Selection of Immune-Related and Inflammatory Markers

Based on the information obtained from reference studies [15,18,32,34,41,42,43,44,45,46,48,49], we screened several markers related to malignancy and performed preliminary tests at the RNA level through quantitative PCR (qPCR). After conducting the qPCR analysis, we selected the following 12 markers that exhibited high expression levels and decided to include them in our research: APRIL/TNFSF13, BAFF, LAG-3, PD-1, PD-L1, CTLA-4, CHIT, MMP-3, osteocalcin, pentraxin-3, sTNF-R1, and sTNF-R2.

### 4.4. Bio-Plex Multiplex Immunoassay System

The concentrations of immune-related markers (APRIL/TNFSF13, BAFF, LAG-3, PD-1, PD-L1, and CTLA-4) and inflammatory markers (CHIT, MMP-3, osteocalcin, pentraxin-3, sTNF-R1, and sTNF-R2) in colon tumor and non-tumor tissues were measured using the Milliplex^®^ map human immuno-oncology checkpoint protein magnetic bead panel 96-well plate assay (Millipore Corporation, Billerica, MA, USA), which is a Luminex-based multiplex technology. The Bio-Plex assays (Bio-Rad, Hercules, CA, USA) contained standard concentrations of each analyte, and the calculated standard curves allowed for the precise definition of the concentration of the protein of interest. The assay was performed according to the manufacturer’s instructions. A lyophilized cytokine standard was resuspended in standard diluents, and a serial dilution of the standard (30 µL) was performed to generate the standard curves for each protein of interest. The bead mixture was added to the standard or sample, and the plate was incubated overnight (16–18 h) at 4 °C. The sample was washed three times with wash buffer using an automatic washer for magnetic beads. The detection antibody was then added to the plate and incubated at room temperature for 1 h. A streptavidin–phycoerythrin mix was added, and the sample was incubated at room temperature for 30 min. After washing, the assay buffer was added, and the sample was analyzed using the Luminex 200 Bio-Plex instrument (Bio-Rad, Hercules, CA, USA).

### 4.5. Surgery and Pathological Examination

Complete mesocolic excision and total mesorectal excision, which are standard surgical approaches for colon and rectal cancer, respectively, were performed. The excised tissue was immediately transported to the pathology department, where a pathologist extracted normal and tumor tissue sections. Histopathological examination was conducted following standardized guidelines, and the resulting histological data were recorded.

### 4.6. Statistical Analyses

The categorical variables were analyzed using the chi-square test and are described as frequencies and percentages. The Fisher’s exact test was performed if the frequency of the data was <5. The normality of all continuous data was tested using the Shapiro–Wilk test. Continuous variables were analyzed using the Student’s *t*-test and are expressed as mean values and standard deviations. Non-normally distributed data were analyzed using the Mann–Whitney U test and are described as medians and interquartile ranges. DFS was defined as the period from the date of the index surgery to the date of tumor recurrence or death. OS was defined as the period from the date of the index surgery to the date of death. The survival analysis was performed using the Kaplan–Meier curve with the log-rank test. All statistical analyses were performed using R statistical software (version 4.1.0; R Foundation for Statistical Computing, Vienna, Austria). Statistical significance was set at *p* < 0.05.

## 5. Conclusions

In conclusion, this study confirmed that APRIL/TNFSF13, BAFF, and MMP-3 are overexpressed in colorectal tumor tissues and the overexpression of APRIL/TNFSF13, BAFF, and MMP-3 indicates their potential relationship with unfavorable clinicopathological features and poor prognosis of CRC. Further research and validation studies are necessary to establish the clinical utility of these markers as biomarkers for CRC. It is worth noting that biomarker discovery and validation are ongoing processes in medical research, and additional evidence is needed to determine the reliability and accuracy of these markers in clinical practice.

## Figures and Tables

**Figure 1 ijms-24-11579-f001:**
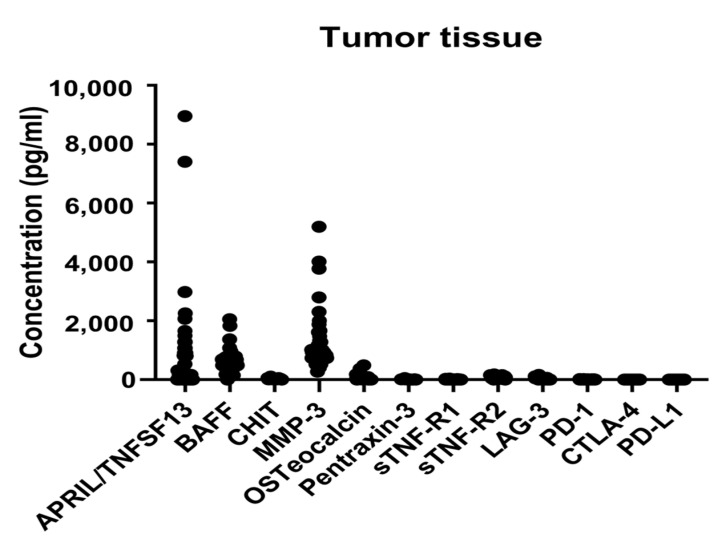
Levels of immune-related and inflammatory markers in the tumor tissues. Scatter plots showing the distribution of the levels of twelve markers. APRIL/TNFSF13, a proliferation-inducing ligand/tumor necrosis factor lsuperfamily member 13; BAFF, B lymphocyte activating factor; CHIT, chitinase 1; MMP-3, matrix metallopeptidase 3; sTNF-R, soluble tumor necrosis factor receptor type; LAG-3, lymphocyte activation gene-3; PD-1, programmed cell death protein 1; PD-L1, programmed death-ligand 1; CTLA-4, cytotoxic T lymphocyte-associated protein 4.

**Figure 2 ijms-24-11579-f002:**
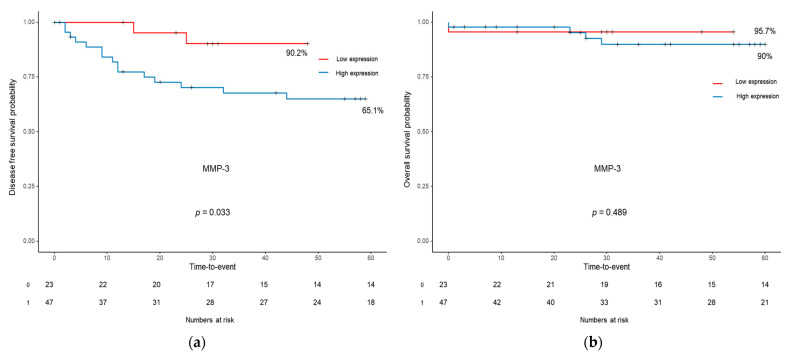
Kaplan–Meier survival curves for disease-free survival (**a**) and overall survival (**b**) according to the expression level of MMP-3. MMP-3, matrix metallopeptidase 3.

**Table 1 ijms-24-11579-t001:** Baseline patient characteristics.

	Number of Patients (n = 70)	Percentage (%)
Age, mean ± SD	69.6 ± 10.8	
Gender		
Male	38	54.3
Female	32	45.7
Body mass index, mean ± SD	23.4 ± 3.5	
ASA score		
II	35	50.0
III	35	50.0
Medical history		
None	19	27.1
One	18	25.7
Two or more	33	47.1
Tumor location		
Right	19	27.1
Left	27	38.6
Rectum	24	34.3
CEA		
<5	45	64.3
≥5	25	35.7
Operation method		
Open	15	21.4
MIS	55	78.6
T stage		
Tis	1	1.4
3	53	75.7
4	16	22.9
N stage		
0	28	40.0
1	28	40.0
2	14	20.0
M stage		
0	57	81.4
1	13	18.6
TNM stage		
0	1	1.4
2	25	35.7
3	31	44.3
4	13	18.6
Metastatic lymph node, mean ± SD	2.2 ± 3.6	
Harvested lymph node, mean ± SD	24.8 ± 11.1	
Tumor differentiation		
Good differentiation	13	18.8
Moderate differentiation	53	76.8
Poor differentiation	1	1.4
Mucinous adenocarcinoma	2	2.9
Tumor size (cm), mean ± SD	5.0 ± 2.0	
Lymphatic invasion		
Negative	38	54.3
Positive	32	45.7
Venous invasion		
Negative	63	90.0
Positive	7	10.0
Perineural invasion		
Negative	50	71.4
Positive	20	28.6
EGFR		
Negative	5	7.6
Positive	61	92.4
MSI		
MSS	63	94.0
MSI-H	4	6.0
KRAS		
Wild	39	58.2
Mutant	28)	41.8
NRAS		
Wild	47	92.2
Mutant	2	3.9
BRAF		
Wild	62	95.4
Mutant	3	4.6
Laboratory markers, mean ± SD		
WBC (10^3^/μL)	7.6 ± 3.1	
Hb (g/dL)	11.9 ± 2.4	
PLT (10^3^/μL)	288.0 ± 100.3	
Neutrophil count (10^3^/μL)	5.4 ± 3.0	
Lymphocyte count (10^3^/μL)	1.5 ± 0.6	
NLR	4.5 ± 4.9	
C-reactive protein (mg/dL)	1.7 ± 2.7	
Albumin (g/dL)	3.8 ± 0.6	
Chemotherapy		
No	24	34.3
Yes	46	65.7
Radiotherapy		
No	69	98.6
Yes	1	1.4
Recurrence		
No	42	60.0
Yes	17	24.3
Death		
No	45	64.3
Yes	5	7.1

SD, standard deviation; ASA, American Society of Anesthesiologists; CEA, carcinoembryonic antigen; MIS, minimally invasive surgery; EGFR, epidermal growth factor receptor; MSI, Microsatellite instability; WBC, white blood cell; Hb, hemoglobin; PLT, platelet; NLR, neutrophil/lymphocyte ratio.

**Table 2 ijms-24-11579-t002:** Immune-related and inflammatory markers in the tumor tissues.

	Median (IQR)	Range
APRIL/TNFSF13	166.02 (0, 806.4)	0~8954.58
BAFF	485.6 (355.3, 664.0)	0~2053.2
CHIT	21.3 (10.24, 31.24)	0~101.19
MMP-3	905.1 (736.2, 1106.9)	270.5~5198.8
Osteocalcin	16.33 (2.37, 41.34)	0~487.48
Pentraxin-3	8.98 (7.41, 12.19)	3.23~57.43
sTNF-R1	6.67 (5.43, 7.87)	2.28~36.20
sTNF-R2	60.99 (35.78, 106.93)	0~177.87
LAG-3	0 (0, 11.46)	0~164.96
PD-1	5.3 (5.30, 10.84)	0~21.77
PD-L1	0 (0, 0.43)	0~4.49
CTLA-4	0 (0, 0)	0~3.1

IQR, interquartile range; APRIL/TNFSF13, a proliferation-inducing ligand/tumor necrosis factor lsuperfamily member 13; BAFF, B lymphocyte activating factor; CHIT, chitinase 1; MMP-3, matrix metallopeptidase 3; sTNF-R, soluble tumor necrosis factor receptor Type; LAG-3, lymphocyte activation gene-3; PD-1, programmed cell death protein 1; PD-L1, programmed death ligand 1; CTLA-4, cytotoxic T lymphocyte-associated protein 4.

**Table 3 ijms-24-11579-t003:** Correlation of immunologic markers with clinicopathologic features.

	APRIL/TNFSF13 (806.4)		*p*	BAFF (664.0)		*p*	MMP-3 (736.2)		*p*
	Low (N = 51)	High (N = 19)		Low (N = 52)	High (N = 18)		Low (N = 23)	High (N = 47)	
Age, mean ± SD	69.8 ± 10.7	69.2 ± 11.0	0.82	69.4 ± 11.1	70.3 ± 10.1	0.74	70.7 ± 11.4	69.1 ± 10.5	0.59
Gender									
Male	27 (52.9)	11 (57.9)	0.92	25 (48.1)	13 (72.2)	0.13	15 (65.2)	23 (48.9)	0.30
Female	24 (47.1)	8 (42.1)		27 (51.9)	5 (27.8)		8 (34.8)	24 (51.1)	
BMI	23.5 ± 3.7	23.1 ± 3.1	0.70	23.4 ± 3.7	23.4 ± 3.0	0.96	23.3 ± 3.6	23.4 ± 3.5	0.90
ASA score									
II	28 (54.9)	7 (36.8)	0.28	29 (55.8)	6 (33.3)	0.17	13 (56.5)	22 (46.8)	0.61
III	23 (45.1)	12 (63.2)		23 (44.2)	12 (66.7)		10 (43.5)	25 (53.2)	
Medical history									
None	14 (27.5)	5 (26.3)	0.95 *	12 (23.1)	7 (38.9)	0.22 *	6 (26.1)	13 (27.7)	0.81
One	14 (27.5)	4 (21.1)		16 (30.8)	2 (11.1)		5 (21.7)	13 (27.7)	
T or more	23 (45.1)	10 (52.6)		24 (46.2)	9 (50.0)		12 (52.2)	21 (44.7)	
Tumor location									
Right	15 (29.4)	4 (21.1)	0.65	17 (32.7)	2 (11.1)	0.13 *	8 (34.8)	11 (23.4)	0.51
Left	20 (39.2)	7 (36.8)		20 (38.5)	7 (38.9)		7 (30.4)	20 (42.6)	
Rectum	16 (31.4)	8 (42.1)		15 (28.8)	9 (50.0)		8 (34.8)	16 (34.0)	
CEA									
<5	34 (66.7)	11 (57.9)	0.68	34 (65.4)	11 (61.1)	0.96	19 (82.6)	26 (55.3)	0.04
≥5	17 (33.3)	8 (42.1)		18 (34.6)	7 (38.9)		4 (17.4)	21 (44.7)	
Operation method									
Open	10 (19.6)	5 (26.3)	0.53 *	11 (21.2)	4 (22.2)	1 *	6 (26.1)	9 (19.1)	0.54 *
MIS	41 (80.4)	14 (73.7)		41 (78.8)	14 (77.8)		17 (73.9)	38 (80.9)	
T stage									
Tis	0 (0.0)	1 (5.3)	0.29 *	0 (0.0)	1 (5.6)	0.09 *	0 (0.0)	1 (2.1)	0.01 *
3	40 (78.4)	13 (68.4)		42 (80.8)	11 (61.1)		22 (95.7)	31 (66.0)	
4	11 (21.6)	5 (26.3)		10 (19.2)	6 (33.3)		1 (4.3)	15 (31.9)	
N stage									
0	22 (43.1)	6 (31.6)	0.60 *	20 (38.5)	8 (44.4)	0.81 *	13 (56.5)	15 (31.9)	0.15 *
1	20 (39.2)	8 (42.1)		22 (42.3)	6 (33.3)		6 (26.1)	22 (46.8)	
2	9 (17.6)	5 (26.3)		10 (19.2)	4 (22.2)		4 (17.4)	10 (21.3)	
M stage									
0	45 (88.2)	12 (63.2)	0.03 *	45 (86.5)	12 (66.7)	0.08 *	21 (91.3)	36 (76.6)	0.19 *
1	6 (11.8)	7 (36.8)		7 (13.5)	6 (33.3)		2 (8.7)	11 (23.4)	
TNM stage									
0	0 (0.0)	1 (5.3)	0.02 *	0 (0.0)	1 (5.6)	0.06 *	0 (0.0)	1 (2.1)	0.04 *
2	21 (41.2)	4 (21.1)		19 (36.5)	6 (33.3)		13 (56.5)	12 (25.5)	
3	24 (47.1)	7 (36.8)		26 (50.0)	5 (27.8)		8 (34.8)	23 (48.9)	
4	6 (11.8)	7 (36.8)		7 (13.5)	6 (33.3)		2 (8.7)	11 (23.4)	
Metastatic lymph node	1.8 ± 3.3	3.2 ± 4.3	0.23	2.0 ± 3.3	2.8 ± 4.4	0.45	1.1 ± 1.7	2.7 ± 4.2	0.02
Harvested lymph node	24.4 ± 9.4	25.9 ± 14.9	0.68	25.3 ± 9.8	23.4 ± 14.4	0.62	27.1 ± 13.1	23.7 ± 9.9	0.27
Tumor differentiation									
WD	10 (19.6)	3 (16.7)	0.79 *	9 (17.3)	4 (23.5)	0.63 *	4 (17.4)	9 (19.6)	0.91 *
MD	39 (76.5)	14 (77.8)		41 (78.8)	12 (70.6)		19 (82.6)	34 (73.9)	
PD	1 (2.0)	0 (0.0)		1 (1.9)	0 (0.0)		0 (0.0)	1 (2.2)	
Mucinous	1 (2.0)	1 (5.6)		1 (1.9)	1 (5.9)		0 (0.0)	2 (4.3)	
Tumor size (cm), mean ± SD	4.9 ± 2.2	5.2 ± 1.3	0.50	4.9 ± 2.2	5.1 ± 1.5	0.64	4.8 ± 2.0	5.0 ± 2.1	0.74
Lymphatic invasion									
Negative	28 (54.9)	10 (52.6)	1	28 (53.8)	10 (55.6)	1	11 (47.8)	27 (57.4)	0.61
Positive	23 (45.1)	9 (47.4)		24 (46.2)	8 (44.4)		12 (52.2)	20 (42.6)	
Venous invasion									
Negative	47 (92.2)	16 (84.2)	0.37 *	47 (90.4)	16 (88.9)	1 *	21 (91.3)	42 (89.4)	1 *
Positive	4 (7.8)	3 (15.8)		5 (9.6)	2 (11.1)		2 (8.7)	5 (10.6)	
Perineural invasion									
Negative	40 (78.4)	10 (52.6)	0.06	40 (76.9)	10 (55.6)	0.15	19 (82.6)	31 (66.0)	0.24
Positive	11 (21.6)	9 (47.4)		12 (23.1)	8 (44.4)		4 (17.4)	16 (34.0)	
EGFR									
Negative	1 (2.1)	4 (22.2)	0.01 *	1 (2.0)	4 (25.0)	0.01 *	0 (0.0)	5 (11.6)	0.15 *
Positive	47 (97.9)	14 (77.8)		49 (98.0)	12 (75.0)		23 (100.0)	38 (88.4)	
MSI									
MSS	46 (92.0)	17 (100.0)	0.56 *	47 (92.2)	16 (100.0)	0.56 *	21 (91.3)	42 (95.5)	0.60 *
MSI-H	4 (8.0)	0 (0.0)		4 (7.8)	0 (0.0)		2 (8.7)	2 (4.5)	
KRAS									
Wild	27 (56.2)	12 (63.2)	0.80	29 (59.2)	10 (55.6)	1	12 (57.1)	27 (58.7)	1
Mutant	21 (43.8)	7 (36.8)		20 (40.8)	8 (44.4)		9 (42.9)	19 (41.3)	
NRAS									
Wild	33 (97.1)	14 (93.3)	0.52 *	35 (97.2)	12 (92.3)	0.46 *	16 (100.0)	31 (93.9)	1 *
Mutant	1 (2.9)	1 (6.7)		1 (2.8)	1 (7.7)		0 (0.0)	2 (6.1)	
BRAF									
Wild	45 (95.7)	17 (94.4)	1 *	45 (93.8)	17 (100.0)	0.56 *	19 (95.0)	43 (95.6)	1 *
Mutant	2 (4.3)	1 (5.6)		3 (6.2)	0 (0.0)		1 (5.0)	2 (4.4)	
Laboratory markers, median [IQR]									
WBC (10^3^/μL)	6.6(5.4, 9.2)	7.1 (6.5, 8.8)	0.53	7.2 (5.5, 9.2)	6.7 (5.9, 8.9)	0.83	6.5 (4.9, 7.6)	7.2 (5.9, 9.4)	0.10
Hb (g/dL)	12.6 (10.4, 13.6)	11.1 (9.7, 12.5)	0.13	12.4 (10.2, 13.4)	12.4 (9.8, 13.8)	0.87	12.4 (10.1, 13.8)	12.3 (10.2, 13.6)	0.58
PLT (10^3^/μL)	272.0 (209.5, 323.0)	253.0 (231.0, 331.0)	0.92	275.5 (212.2, 333.5)	242.0 (224.5, 294.2)	0.37	260.0 (193.0, 307.0)	259.0 (222.5, 332.5)	0.42
Neutrophil (10^3^/μL)	4.7 (3.0, 6.4)	5.1 (4.4, 7.1)	0.17	4.7 (3.1, 6.9)	4.9 (4.3, 6.8)	0.38	3.6 (3.0, 5.8)	4.9 (3.7, 7.1)	0.08
Lymphocyte (10^3^/μL)	1.6 (1.3, 1.9)	1.3 (1.0, 1.8)	0.20	1.5 (1.2, 1.9)	1.3 (1.0, 1.8)	0.49	1.4 (1.2, 1.7)	1.6 (1.1, 1.9)	0.48
NLR	2.7 (2.1, 4.2)	4.1 (2.7, 6.0)	0.04	2.7 (2.2, 4.4)	3.9 (2.7, 5.4)	0.15	2.5 (2.1, 4.1)	3.6 (.5, 5.2)	0.16
CRP (mg/dL)	0.4 (0.3, 1.6)	1.0 (0.3, 2.3)	0.25	0.5 (0.3, 1.8)	0.7 (0.3, 1.3)	0.97	0.6 (0.3, 1.3)	0.7 (0.3, 1.8)	0.99
Albumin (g/dL)	3.9 (3.6, 4.3)	3.7 (3.2, 4.0)	0.07	3.9 (3.6, 4.2)	3.8 (3.3, 4.2)	0.58	3.8 (3.2, 4.2)	3.9 (3.5, 4.3	0.31
Chemotherapy									
No	17 (33.3)	7 (36.8)	1	16 (30.8)	8 (44.4)	0.44	9 (39.1)	15 (31.9)	0.74
Yes	34 (66.7)	12 (63.2)		36 (69.2)	10 (55.6)		14 (60.9)	32 (68.1)	
Radiotherapy									
No	50 (98.0)	19 (100.0)	1 *	51 (98.1)	18 (100.0)	1 *	22 (95.7)	47 (100.0)	0.32 *
Yes	1 (2.0)	0 (0.0)		1 (1.9)	0 (0.0)		1 (4.3)	0 (0.0)	
Recurrence									
No	34 (66.7)	8 (42.1)	0.06 *	35 (67.3)	7 (38.9)	0.03 *	16 (69.6)	26 (55.3)	0.08 *
Yes	12 (23.5)	5 (26.3)		12 (23.1)	5 (27.8)		2 (8.7)	15 (31.9)	
Death									
No	36 (70.6)	9 (47.4)	0.11 *	37 (71.2)	8 (44.4)	0.08 *	16 (69.6)	29 (61.7)	0.78 *
Yes	4 (7.8)	1 (5.3)		3 (5.8)	2 (11.1)		1 (4.3)	4 (8.5)	

SD, standard deviation; ASA, American Society of Anesthesiologists; CEA, carcinoembryonic antigen; MIS, minimally invasive surgery; EGFR, epidermal growth factor receptor; MSI, microsatellite instability; IQR, interquartile range; WBC, white blood cell; Hb, hemoglobin; PLT, platelet; NLR, neutrophil/lymphocyte ratio. * Statistical analysis for this variable was performed using Fisher’s exact test.

## Data Availability

Not applicable.

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
