# Peer review of "Expression of Immune-Related and Inflammatory Markers and Their Prognostic Impact in Colorectal Cancer Patients"

_ijms, 2023, doi:10.3390/ijms241411579_

Round 1

Reviewer 1 Report

In this paper, the authors identified immune-related and inflammatory markers expressed in Colorectal cancer tissues and investigated whether they were related to clinicopathological features and prognosis. The introduction provides enough information for the reader to understand the subject very clear aim. Methods are very clearly described. Results are quite descriptive and bring all relevant information in details. The discussion is well written with relevant association with studies in the literature and also presents the main results and their meaning in a clear way. Therefore, I recommend the publication of the manuscript after very minor corrections, some of which I described below:

Page 10, line 169

replace was by were.

Rewrite the first paragraph of the discussion. It can be combined so the words furthermore, in addition and moreover are not used in every sentence.

Page 12, lines 241-245.

The last sentence is too isolated. I would suggest to rewrite it like:

“The mulitplex immunoassay has been used in a few studies to detect tumor-specific biomarkers in malignant tumors such as melanomas, ovarian cancer, pancreatic cancer [37,38], and CRC, used, for example, by Calu et al.[13] to identify inflammatory molecules.”

Some sentences should be rewritten to improve coherence and the use of repetitions within paragraphs. verbs in singular and plural must be revised as well.

Author Response

Comment 1: Page 10, line 169, replace was by were.

Reviewed and Corrected: The authors appreciate the reviewer for their careful reviews.  As per your suggestion, was replaced into were in line-169, page 10, and highlighted with red color in our newly revised manuscript.

Comment 2: Rewrite the first paragraph of the discussion. It can be combined so the words furthermore, in addition, and moreover are not used in every sentence.

Reviewed and Corrected: The authors appreciate the reviewer for their careful reviews. As per your comments, we revised the first paragraph of the discussion section in our newly revised manuscript with highlighted red color.

“In this study, we investigated the expression of immune-related and inflammatory markers in CRC tissue samples. The results indicated that APRIL/TNFSF13, BAFF, and MMP-3 are highly expressed in CRCs, and high expression of immune-related and inflammatory markers were associated with advanced clinicopathological features. In our analysis, high expression of MMP-3 was associated with elevated CEA levels, more lymph node metastases, and more advanced disease stages, and MMP-3 expression level was associated with long-term prognoses, such as five-year DFS. In addition, the high APRIL/TNFSH13 expression group showed a higher rate of metastatic lesions than the low APRIL/TNFSH13 expression group.”

Comment 3: Page 12, lines 241-245. The last sentence is too isolated. I would suggest to rewrite it like: The last sentence is too isolated. I would suggest to rewrite it like:

“The multiplex immunoassay has been used in a few studies to detect tumor-specific biomarkers in malignant tumors such as melanomas, ovarian cancer, pancreatic cancer [37,38], and CRC, used, for example, by Calu et al.[13] to identify inflammatory molecules.”

Reviewed and Corrected: The authors appreciate the reviewer for their careful reviews.  As per your suggestion, we have written this sentence that you mentioned in our newly revised manuscript with highlighted yellow color.

“The multiplex immunoassay has been used in a few studies to detect tumor-specific biomarkers in malignant tumors such as melanomas, ovarian cancer, and pancreatic cancer [40,41], in which, Calu et al. [15], used multiplex immunoassays to identify inflammatory molecules in CRC tissues.”

Comment 4: Some sentences should be rewritten to improve coherence and the use of repetitions within paragraphs. verbs in singular and plural must be revised as well.

Reviewed and Corrected: The authors appreciate the reviewer for their careful reviews.  As per your suggestion, we have improved coherence and the use of repetitions within paragraphs throughout the manuscript as well as grammatical errors.

Reviewer 2 Report

In the manuscript entitled “Expression of immune-related and inflammatory markers and their prognostic impact in colorectal cancer patients”, the authors explored the expression of immune-related and inflammatory markers in CRC tissue samples, and found that the APRIL/TNFSF13, BAFF, and MMP-3 pathway is overexpressed in CRC tissues and is associated with unfavorable clinicopathological features and poor prognosis in CRC patientsit. These markers may serve as potential biomarkers for the diagnosis or prognosis of CRC. It has important clinical significance, however there are some concerns about the manuscript.

1.     In this paper, the OS and five-year DFS of patients with differences in APRIL/TNFSH13, BAFF and MMP-3 markers were statistically analyzed. Is there any comparison of their PFS (Progression-Free Survival)?

2.     The introduction section could introduce more relevant factors that can be used as immune markers in CRC that have been reported thus far;

3.     In the discussion, it is reported that APRIL/TNFSF13, BAFF, and MMP-3 pathways seem to be related to tumor invasion and metastasis. Perhaps you can focus on the relationship between these factors and CRC invasion and metastasis in the future;

4.     The manuscript has a small number of grammatical errors, inappropriate words or missing punctuation that should be fixed before it may be found fit for publication.

Good.

Author Response

Comment 1.:  In this paper, the OS and five-year DFS of patients with differences in APRIL/TNFSH13, BAFF and MMP-3 markers were statistically analyzed. Is there any comparison of their PFS (Progression-Free Survival)?

Reviewed and Corrected: The authors appreciate the reviewer for their careful reviews. The authors understood the meaning of your comment. Our study included stage 2, 3, and 4 colorectal cancer patients, who underwent radical resection with curative intent. We considered the progression of stage 4 patients as recurring and analyzed disease-free survival. Therefore, I think the PFS you mentioned can be seen as the same as the DFS in our study.

*Disease-Free Survival (DFS), also known as Relapse-Free Survival (RFS), is often used as a primary endpoint in phase III trials of adjuvant therapy. Progression-Free Survival (PFS) is commonly used as a primary endpoint in phase III trials evaluating the treatment of metastatic cancer.

Comment 2.: The introduction section could introduce more relevant factors that can be used as immune markers in CRC that have been reported thus far;

Reviewed and Corrected: The authors appreciate the reviewer for their careful reviews.  As per your suggestions, we have added several relevant factors in the introduction section that can be used as immune markers in CRC that have been reported thus far in our newly revised manuscript with highlighted red color.

"In recent years, there has been research focused on identifying and targeting additional immune checkpoints in the tumor microenvironment to enhance the effectiveness of immunotherapy. The immune checkpoints, including lymphocyte activation gene-3 (LAG-3), T cell immunoglobulin and mucin domain 3 (TIM-3), B7-homolog 3 (B7-H3), V-domain immunoglobulin-containing suppressor of T-cell activation (VISTA), diacylglycerol kinase-α (DGK-α), T cell immunoglobulin and ITIM domain (TIGIT), and B and T lymphocyte attenuator (BTLA), have gained attention as potential targets for immunotherapy [11]. Cytotoxic lymphocytes, particularly cytotoxic T lymphocytes (CTLs), are considered crucial components of the immune system's anti-tumor response. CTLs play a vital role in recognizing and eliminating cancer cells [12]."

References

Mehdizadeh, S.; Bayatipoor, H.; Pashangzadeh, S.; Jafarpour, R.; Shojaei, Z.; Motallebnezhad, M. Immune checkpoints and cancer development: Therapeutic implications and future directions. Pathology-Research and Practice 2021, 223, 153485.

Deschoolmeester, V.; Baay, M.; Lardon, F.; Pauwels, P.; Peeters, M. Immune cells in colorectal cancer: prognostic relevance and role of MSI. Cancer microenvironment 2011, 4, 377-392.

Comment 3.:   In the discussion, it is reported that APRIL/TNFSF13, BAFF, and MMP-3 pathways seem to be related to tumor invasion and metastasis. Perhaps you can focus on the relationship between these factors and CRC invasion and metastasis in the future.

Reviewed and Corrected: The authors appreciate the reviewer for their careful reviews.  Yes, the authors agree with your opinion and suggestions. In the future, we have planned to study related to this with samples from a more significant number of patients.

Comment 4.:  The manuscript has a small number of grammatical errors, inappropriate words or missing punctuation that should be fixed before it may be found fit for publication.

Reviewed and Corrected: The authors appreciate the reviewer for their careful reviews. As per your suggestions, the authors carefully checked all inappropriate words, missing punctuation, and grammatical errors throughout the revised manuscript and corrected them.

Reviewer 3 Report

In the manuscript presented by An et al., the expression of 6 immuno-related markers and 6 inflammatory markers is evaluated in samples from patients with colorectal cancer. Although the authors mention that the APRIL/TNFS13, BAFF and MMP-3 markers are overexpressed in CRC tumor tissue compared to healthy tissue, the experimental design and justification for evaluating only 6 immuno-related markers and 6 inflammatory markers it is not explained anywhere in the text, leaving more doubts than certainties.

A screening of several markers could have been carried out using a microarray to determine the overexpression or subexpression of these and other markers and, later, corroborate their expression using the bio-plex multiplex immunoassay system.

In this same sense, only in discussion is the function of the 3 overexpressed markers briefly mentioned, but the functions of the other evaluated markers are not mentioned.

In section 2.3 of results it is mentioned that the markers were grouped in certain quartiles (lines 127-128), but it is not explained why it was decided to group them in this way.

Table 3: what does the asterisk mean?

Table 1: in the same column, average values are being combined with frequencies and percentages. Separate it

In conclusions (lines 317-318): it would be more accurate to say that the markers APRIL/TNFSF13, BAFF, and MMP-3 were overexpressed, excluding "various"

Author Response

Comments 1: The experimental design and justification for evaluating only 6 immuno-related markers and 6 inflammatory markers it is not explained anywhere in the text, leaving more doubts than certainties.

Comments 2: In this same sense, only in discussion is the function of the 3 overexpressed markers briefly mentioned, but the functions of the other evaluated markers are not mentioned.

Reviewed and Corrected: Thank you for your valuable feedback and insightful question regarding our choice of inflammatory markers. We appreciate the opportunity to address your query and provide further clarification on our selection process. Inflammation is known to be associated with the immune response. However, to our knowledge, there have been no studies that have simultaneously investigated inflammatory markers and immune markers in tumor tissue. Therefore, we tried to simultaneously examine the expression patterns of immune markers and inflammatory markers in the same samples.

In our study, we carefully considered several factors when choosing the inflammatory markers for investigation. These factors included the existing literature on colorectal cancer and inflammation, the relevance of the markers to the pathophysiology of colorectal cancer, and the availability of reliable assays to measure these markers accurately. Based on the information obtained from reference studies, we screened various kinds of markers related to malignancy and performed preliminary tests at the RNA level through qPCR. Subsequently, we selected promising markers and conducted our main research using Bioplex immunoassay.

Allow us to explain our rationale behind selecting each marker:

  1. MMP3 (Matrix Metalloproteinase-3): MMP3 is a well-studied member of the matrix metalloproteinase family that plays a crucial role in extracellular matrix remodeling and tumor invasion. Numerous studies have linked MMP3 expression to tumor progression and metastasis in colorectal cancer specifically. By including MMP3, we aimed to assess its potential as a prognostic marker for poor outcomes in these patients.
  2. CHIT (Chitotriosidase): CHIT is an enzyme involved in the breakdown of chitin, a component of fungal cell walls. Recent research has highlighted the role of CHIT in modulating inflammation and its potential association with cancer. Although its exact mechanism in colorectal cancer is not yet fully understood, we included CHIT to explore its potential predictive value in assessing patient outcomes.
  3. Osteocalcin: While primarily recognized as a marker of bone metabolism, emerging evidence suggests a potential link between osteocalcin and cancer, including colorectal cancer. Osteocalcin has been implicated in modulating tumor growth, angiogenesis, and metastasis. Given its relevance in cancer biology, we incorporated osteocalcin in our study to determine its utility as a predictor of poor outcomes in colorectal cancer patients.
  4. Pentraxin-3: Pentraxin-3 is an acute-phase protein involved in the innate immune response and inflammation. Several studies have implicated pentraxin-3 in colorectal cancer progression, tumor invasiveness, and metastasis. By including pentraxin-3, we aimed to evaluate its potential as an inflammatory marker for identifying patients at a higher risk of adverse outcomes in colorectal cancer.
  5. sTNF-R1 and sTNF-R2 (soluble Tumor Necrosis Factor Receptors 1 and 2): Tumor necrosis factor (TNF) is a pro-inflammatory cytokine involved in cancer-related inflammation and immune response. The soluble forms of TNF receptors, sTNF-R1 and sTNF-R2, have been investigated as potential biomarkers in various cancers. In colorectal cancer, altered levels of these receptors have been associated with disease progression and poor prognosis. We selected sTNF-R1 and sTNF-R2 to evaluate their potential as predictive markers for poor outcomes in our study.

In conclusion, our selection of these specific inflammatory markers was based on their relevance to the pathophysiology of colorectal cancer, the existing literature supporting their association with poor outcomes, and the availability of reliable assays for their measurement. We believe that the inclusion of these markers will contribute to a comprehensive assessment of their effectiveness in predicting adverse outcomes in colorectal cancer patients.

Thank you again for your valuable input, which has allowed us to provide further clarification on our choice of inflammatory markers. We hope this response adequately addresses your concerns and contributes to the improvement of our manuscript. We summarized the contents related to this and briefly added them to the method part and discussion part.

Comments 3: In section 2.3 of results it is mentioned that the markers were grouped in certain quartiles (lines 127-128), but it is not explained why it was decided to group them in this way.

Reviewed and Corrected: Thank you so much for your crucial and constructive comments. We set a quartile value that can be used as a representative value as a reference point. Since the absolute expression value or cut-off value of each marker has not been established so far, we set the cut-off value as a representative value showing clinically meaningful results for each marker. In this respect, we thought that there may be limitations in generalizing the results of our study and added this to the limitation part of the discussion.

Comments 4.: Table 3: what does the asterisk mean?

Reviewed and corrected: The authors appreciate the reviewer for their careful reviews. The asterisk (*) indicates that the results were analyzed using Fisher's exact test. I have added this note below Table 3.

Comments 5.: Table 1: in the same column, average values are being combined with frequencies and percentages. Separate it.

Reviewed and corrected: The authors appreciate the reviewer for their careful reviews. We have made the modifications in Table 1 and separated one more for % in our newly revised manuscript with highlighted red color.

Comments 6.: In conclusions (lines 317-318): it would be more accurate to say that the markers APRIL/TNFSF13, BAFF, and MMP-3 were overexpressed, excluding "various"

Reviewed and corrected: The authors appreciate the reviewer for their careful reviews. As per your suggestions, it has been revised in our newly revised manuscript with highlighted red color.

"In conclusion, this study confirmed that APRIL/TNFSF13, BAFF, and MMP-3 are overexpressed in colorectal tumor tissues and the overexpression of APRIL/TNFSF13, BAFF, and MMP-3 indicates their potential relationship with the unfavorable clinicopathological features and poor prognosis of CRC. Further research and validation studies would be necessary to establish the clinical utility of these markers as biomarkers for CRC. It's worth noting that biomarker discovery and validation are ongoing processes in medical research, and additional evidence is needed to determine the reliability and accuracy of these markers in clinical practice."

Round 2

Reviewer 3 Report

Thanks to the authors for address the comments. I have not more suggestions